# Effects of a Comprehensive Person-Centered Care Education Program for Nursing Students

**DOI:** 10.3390/medicina59030463

**Published:** 2023-02-25

**Authors:** Myoungsuk Kim

**Affiliations:** College of Nursing, Kangwon National University, Chuncheon 24341, Republic of Korea; cellylife@gmail.com

**Keywords:** nursing students, patient-centered care, education, empathy, communication

## Abstract

*Background and Objectives*: The aim of this study is to develop and implement a comprehensive person-centered care (PCC) education program for nursing students and assess its effects on individualized care, empathy, communication competence, and clinical practice stress. *Materials and Methods*: A non-equivalent control group non-synchronized design was used. Participants were 60 third-year nursing students undergoing clinical practicum in a nursing school. They were assigned either to the control group (29 students) or experimental group (31 students). The experimental group participated in a comprehensive PCC education program, while the control group did not. Four 65-min sessions were conducted over two weeks; each session comprised 5 min of introduction, 50 min of education, and 10 min of sharing of thoughts and training. Individualized care, empathy, communication competence, and clinical practice stress were measured. Data were collected immediately before the education program and two weeks after program completion. *Results*: After the education program, the experimental group showed significant improvements in individualized care, empathy, and communication competence and significantly reduced clinical practice stress compared to the control group. *Conclusions*: The comprehensive PCC education program is a potentially beneficial intervention for nursing students to help them practice person-centered care with confidence. Longitudinal randomized controlled trials are needed to substantiate these findings.

## 1. Introduction

The paradigm of global healthcare is shifting from a disease-centered to a person-centered perspective. Person-centered care (PCC) refers to a type of care that considers individuals’ values and preferences, informs the patient about all aspects of health management, and supports realistic health and life goals [1]. In other words, it is a type of care that focuses on the patient, respects the patient, actively responds to the patient, and ensures the patient’s rights and autonomy in treatment decisions [2]. Essential elements of PCC are individual focus, empathy, communication, respect, and coordinated care [1,3]. Patients who receive PCC feel more respected through individualized care, develop more confidence in improving their own health, and feel that they are given more support [4]. PCC enhances patients’ health-related quality of life by reducing chronic disease morbidity and improving lifestyle [5]. It also improves nursing care providers’ job satisfaction [6] and quality of care [7]. Thus, PCC is a core competency that must be continually advanced. 

Nursing students become partially involved in patient care during their clinical practicum [8]; therefore, it is imperative that PCC competence is cultivated through effective education programs so that nursing students who will be prospective nurses can provide quality patient care. Existing studies on nursing students conducted abroad have focused on PCC learning experiences for a semester [9] and introduction to teaching approaches [10,11]; studies developing PCC education programs are scarce. In Korea, one study showed that a PCC education program based on a design-thinking approach improved the perceptions of individualized care [8], but most of the published studies investigated the predictors of PCC competence in nursing students [12,13,14]. Although not PCC education programs for nursing students, a prior study implemented a person-centered dementia education program with nursing care providers in long-term care facilities [15], while another applied a PCC education program among clinical nurses [16]. However, research on developing a PCC education program with practical applications for nursing students and evaluating its effectiveness has been insufficient. 

An effective educational approach is critical to enhance nursing students’ levels of caring [9,17]. Furthermore, as PCC education focused on case studies helps nursing students to gain an understanding of PCC fundamentals and broadens their view of PCC [9], case study-based education is as important as theoretical education in cultivating knowledge and attitudes to promote PCC practice. This study utilizes case studies that would help students apply their learned knowledge in practice as well as theoretical education for PCC. Nursing students must also be taught effective communication skills to build an empathetic relationship with their patients to provide PCC [2,18,19]. In other words, carefully listening to and communicating with patients is crucial to accurately understand their motivations, priorities, and preferences [1]. Empathy is also an important factor for practicing PCC [3]. It is important in providing meaningful care, which enhances patients’ well-being [20]. Thus, PCC education for nursing students must also address empathy as well as communication skills. 

No study has yet reported the effects of a comprehensive PCC education program that encompasses a theoretical approach, PCC cases that help apply knowledge in practice, and education to boost communication competence and empathy. Therefore, this study developed a comprehensive PCC education program for nursing students undergoing clinical practicum and investigated the effects of the program on the key elements of PCC—individualized care, empathy, and communication competence—and clinical practice stress. 

## 2. Methods

### 2.1. Study Design

This study used a non-equivalent control group non-synchronized design. Conducting the study simultaneously with the experimental and control groups would have elevated the risk of diffusion effect, as participants were undergoing clinical practicum at the same hospital. Therefore, this study used a non-synchronized design, wherein the control group underwent the study first, followed by the experimental group.

### 2.2. Participants

The inclusion criteria were (1) third-year nursing students who had experienced clinical training for more than one semester at a general hospital, (2) scheduled for clinical practicum during the study period, and (3) no prior participation in a similar PCC education program. The exclusion criteria were (1) unavailable to undergo at least two weeks of on-campus practicum and clinical practicum in the hospital after the completion of the education program and (2) unable to participate in all sessions of the education program.

Sample size was calculated using the G*Power 3.1 software. For independent t-tests with a significance (α) of 0.05, power of 0.8, and effect size of 0.8, the minimum sample size was 26 for each group. The dropout criteria were missing any of the four sessions of the program or failure to complete the pre-test or post-test. 

### 2.3. Enrollment, Allocation, and Blinding

Participants were recruited among third-year nursing students at a nursing school in C city in South Korea through social network service and bulletin board. The researcher anticipated a high dropout rate as the study was conducted during clinical practicum. Considering about 30% dropout, the researcher intended to recruit 68 participants, but 73 participants were enrolled. The recruiting process, educational intervention, and surveys were conducted from 18 May 2021 to 10 December 2021. 

A third person not involved in the study arbitrarily assigned participants into the experimental group and control group. Participants were given a detailed explanation about the study procedure during the informed consent process; they were aware of which program they were participating in. The researcher administered the education program; therefore, the researcher was also aware of the provided content. To reduce the consequent bias by these problems, this study used a non-synchronized design to ensure that the control group and experimental group did not have any contact with each other, and the researcher followed a manual when providing education to ensure consistency in each session.

The experimental group completed the pre-test immediately before beginning the education program and completed the post-test after two weeks of on-campus and clinical practicum following program completion. The control group also completed the post-test after two weeks of on-campus and clinical practicum following the pre-test. The researcher and research assistants administered the pre-test and post-test.

Of 73 participants, 6 participants who could not undergo at least 2 weeks of on-campus and clinical practicum following the completion of the education program were excluded. Of 67 participants meeting the criteria, the control group was expected to have a higher dropout rate because they only complete the pre-test and post-test without intervention; therefore, 36 participants were assigned to the control group and 31 were assigned to the experimental group. A total of 7 participants in the control group failed to complete the post-test, resulting in a total of 29 participants, and all 31 participants in the experimental group were included. Thus, a total of 60 participants were included in the final analysis (Figure 1).

### 2.4. Development Process of Comprehensive PCC Education Program

Existing studies were reviewed to identify the essential contents for the comprehensive PCC education program. Nursing students need to know the concept and principles of PCC to perform PCC during clinical practicum. Thus, this study reviewed previous studies [1,2,3,21]. Detailed PCC education that provides practical assistance is needed for nursing students, and case studies help students understand PCC fundamentals [9]. Thus, this study included PCC cases proposed by Price [4] that can be applied in clinical practicum. 

Moreover, empathy, communication, respect, relationship, and individualized focus are important factors in PCC [3]. This study examined previous research on improving empathy and communication training in nursing students [3,22,23].

The program was structured to include the definition of PCC, core premises, and effects of PCC to help students gain an understanding of PCC, broaden their view, and cultivate a positive attitude toward PCC. Case studies of PCC were taken from the book written by Price [4] and translated into Korean. Three cases were included: helping a patient understand their situation (patient with multiple sclerosis), helping a patient learn about the treatment, care, and recovery (pediatric patient with diabetes), and helping a patient cope with changes and anxiety (patient with colorectal cancer).

Empathy training included contents about understanding the concept of empathy, knowing one’s emotions, expressing emotions, and listening. The goal of the empathy training was to ensure that the students had an accurate understanding of empathy, knew how to express empathy, recognized their own emotions, and listened correctly to identify others’ emotions. Communication training comprised open-ended questions, affirmations, I-message, and reflective listening. The goal of communication training was to help nursing students to build a relationship of trust with the patient, identify the patient’s values, needs, and preferences, deliver information individually, and help patients actively improve their own lifestyle. 

This study set the program to 4 sessions with reference to the 2–6 sessions used by a study providing a person-centered dementia care education program to nursing care providers in long-term care facilities [15] and a study providing a PCC education program to clinical nurses [16]. Each session was set to last 65 min based on the 60min session used previously [16], and each session comprised an introduction (5 min), education (50 min), and hands-on practice and sharing of thoughts (10 min). The educational approaches used were lectures, role play, hands-on practice, and sharing of thoughts based on previous studies [16]. The study parameters were assessed before and after the program using self-report questionnaires. The questionnaires were administered at baseline and after two weeks of on-campus and clinical practicum following the intervention. 

The educational materials were written in 12-point font size to provide to the participants at each session. Moreover, a training manual for each session was developed to ensure consistency in the education program. The lectures consisted of 20 min of understanding of PCC and case study analysis, 15 min of empathy training, and 15 min of communication training for each session (Table 1).

The content validity of the developed comprehensive PCC education program was evaluated. The content on understanding of PCC and the case studies were reviewed by a professor with experience in PCC research and two nursing students, and the empathy and communication training contents were reviewed by three professors with experience in administering relevant education and two nursing students. 

### 2.5. Intervention

The developed comprehensive PCC education program consisted of a total of four sessions (two sessions per week) for 65 min per session. The researcher conducted the program in a quiet lecture room. The students were divided into three-person or four-person groups to promote interaction.

In the initial part of the program, the researcher introduced the outlines for the corresponding session and had students share their thoughts on the changes they had during clinical practicum since the preceding session. During hands-on training, the students could practice what they learned during the education, and during sharing of thoughts, they shared their thoughts after each session. Both lecture and practice were included in each session to enhance students’ understanding and utilization of educational contents. In addition, the researcher tried to identify and solve students’ difficulties during practice. The researcher asked students to think about and express what appropriate empathy and communication should be provided to patients and how to address situations through a case study. However, whether the students understood the contents of each session well was not objectively evaluated in each session. 

The lectures were given according to the manual, and the students were given educational materials and a book containing PCC cases. The education program was administered by the researcher, who has experience in research on communication-enhancing programs and PCC and has translated a book on PCC.

### 2.6. Measurements

Participants’ sex, age, religion, health status, satisfaction with college, satisfaction with nursing major, and academic performance in the preceding semester were surveyed. Individualized care, empathy, communication competence, and clinical practice stress were measured immediately before the intervention and two weeks after the intervention using the self-report questionnaires. The pre-test was written based on students’ experience of providing care to patients during clinical practice before the comprehensive PCC education program, and the post-test was written based on the experience of providing care to patients during the on-campus practicum and the clinical practicum in the hospital after the completion of the education program.

Individualized care was assessed using the Individualized Care Scale-Nurse A version (ICS-A-Nurse) developed by Suhonen et al. [24] and adapted to Korean for use among nursing students by Park [25]. The Individualized Care Scale measures individualized patient care based on nurses’ perceptions and includes the following: (1) assessment of patients’ needs, preferences, and perceptions; (2) patients’ participation in their care; and (3) care based on patient’s individualized information [24], which are key elements of person-centered care. This 17-item instrument uses a 5-point Likert scale, and a higher score indicates greater individualized care. The Cronbach’s α was 0.89 in the study by Park [25] and 0.87 in this study. 

Empathy was assessed using the Jefferson Scale of Empathy-Health Profession Students’ version (JSE-HPS) developed by Hojat et al. [26] and adapted to Korean and validated by Hong [27]. Permission to use the Korean version of the instrument was obtained from Thomas Jefferson University. This instrument contains 20 items (10 negatively worded, 10 positively worded) and uses a 7-point Likert scale, wherein a higher score indicates greater empathy. The Cronbach’s α was 0.88 in the study by Hong [27] and 0.85 in this study. 

Communication competence was measured using the Global Interpersonal Communication Competence (GICC) scale developed by Hur [28]. This 15-item instrument uses a 5-point Likert scale, and a higher score indicates better communication competence. The Cronbach’s α was 0.72 in the study by Hur [28] and 0.80 in this study.

Clinical practice stress was assessed using the instrument developed by Beck and Srivastava [29] and adapted to Korean and modified by Lee and Kim [30]. This 24-item instrument uses a 5-point Likert scale, and a higher score indicates greater clinical practice stress. The Cronbach’s α was 0.91 in the study by Lee and Kim [30] and 0.89 in this study.

### 2.7. Data Analysis

The collected data were analyzed using SPSS version 25.0 (IBM Corp., Armonk, NY, USA). The baseline homogeneity of general characteristics between the experimental and control groups was tested using the chi-squared test, Fisher’s exact test, and independent t-test, and the normality of the dependent variables was analyzed with the Shapiro–Wilk test. The baseline homogeneity of the dependent variables and the effects of the comprehensive PCC education program were analyzed using the Mann–Whitney test or independent t-test depending on the normality of the data. 

### 2.8. Ethical Considerations

This study was approved by the Ethics Committee of Kangwon National University (IRB No.: KWNUIRB-2021-04-011-001) prior to data collection and intervention administration. All participants volunteered to participate in the study and were provided detailed explanations about the purpose and procedure of study, guarantee of anonymity of collected data, benefits and risks, confidentiality, and withdrawal from the study both verbally and in writing. The study began after obtaining written informed consent. 

## 3. Results

### 3.1. Test of Homogeneity for the General Characteristics of Subjects

There were no significant differences in sex, age, religion, subjective health status, satisfaction with college life, satisfaction with nursing major, and academic performance in the preceding semester between the two groups, confirming baseline homogeneity in the general characteristics between the two groups (Table 2).

### 3.2. Baseline Homogeneity of the Dependent Variables

With the exception of individualized care (experimental group W = 0.964, *p* = 0.363, control group W = 0.919, *p* = 0.029), all dependent variables, namely empathy (experimental group W = 0.945, *p* = 0.116, control group W = 0.933, *p* = 0.067), communication competence (experimental group W = 0.969, *p* = 0.500, control group W = 0.987, *p* = 0.965), and clinical practice stress (experimental group W = 0.948, *p* = 0.139, control group W = 0.971, *p* = 0.579), were normally distributed. There were no significant differences in these dependent variables between the two groups, confirming baseline homogeneity (Table 3).

### 3.3. Effects of Comprehensive PCC Education Program

Table 4 shows the changes after the comprehensive PCC education program. The changes in individualized care, empathy, communication competence, and clinical practice stress scores after the education program significantly differed between the experimental group and the control group. Compared to the control group, individualized care (Z = −2.44, *p* = 0.024), empathy (t = −2.76, *p* =0.008), and communication competence (t = −3.25, *p* = 0.002) were significantly improved. Individualized care scores increased in the experimental group (5.61 ± 11.38) but decreased in the control group (−0.24 ± 6.26). Empathy scores increased in the experimental group (3.29 ± 6.90) but decreased in the control group (−2.96 ± 4.48). Additionally, communication competence scores in the experimental group increased (0.77 ± 4.42) but decreased in the control group (−2.96 ± 4.48). Finally, clinical practice stress (t = 2.90, *p* = 0.005) was significantly reduced in the experimental group compared to the control group. Clinical practice stress scores decreased in the experimental group (−1.41 ± 7.93) but increased in the control group (4.48 ± 7.81) (Table 4).

## 4. Discussion

This study aimed to evaluate the effects of a comprehensive PCC education program on nursing students undergoing clinical practicum in the hospital. No study has yet reported the effectiveness of a comprehensive PCC education program for nursing students in Korea and abroad. Therefore, this study is significant as the first study investigating the effects of a comprehensive PCC education program on individualized care, empathy, communication competence, and clinical practice stress among nursing students undergoing clinical practicum.

First, the comprehensive PCC education program significantly improved individualized care compared to the control group. We could not compare our results directly with those in the literature since the study that applied flipped and simulated learning to enhance nursing students’ understanding of PCC [11]. However, it is consistent with the study that showed individualized care improved after an education program based on a design-thinking approach in nursing students [8]. In our study, there are several reasons why the comprehensive PCC education program was effective. First, the fact that the comprehensive PCC education program in this study was effective in enhancing individualized care may be attributable to case study-based education. A theoretical approach to PCC alone does not adequately help nursing students to understand PCC in nursing practice and consider their personal views toward PCC [9]. Thus, we speculate that including case-based learning in PCC education for nursing students was effective. Furthermore, the program simultaneously improved empathy and communication competence, which are the essential factors of PCC [3,19], and this is presumed to have contributed to boosting individualized care in nursing students. Finally, the students were given an opportunity to apply what they had learned in the education program during on-campus and clinical practicum for two weeks, which also would have contributed to enhancing individualized care.

Second, the comprehensive PCC education program also significantly enhanced empathy compared to the control group, and empathy is an essential attribute for nursing students to promote PCC, owing to its strong association with PCC competence [14]. Understanding the patient’s stance and expressing empathy are key features of PCC [22]. Korean nursing curricula include one semester of a communication course in the first year, and although students learn about empathy as part of this course, it is difficult to enhance empathy only through a short class. Thus, various scenarios of empathy applicable to clinical practicum should be developed to train students. In a systematic review of the literature on empathy education for nursing students [31], the most effective empathy education was indicated to be involved immersive and experiential simulation-based interventions. The results of this study highlight the importance of not only theoretical education about empathy but also its practice directly through case studies.

Third, the comprehensive PCC education program significantly improved communication competence compared to the control group. Effective communication between patients and nurses is an essential requirement for PCC [23]. Among the five aspects of professional nursing competence in nursing students, communication was most strongly associated with PCC competence [14]. In this sense, an improvement of communication skills after completing a comprehensive PCC education program is meaningful. As previously mentioned, nursing students in Korea take a communication course only in their first year of school. However, first-year students lack a deep understanding of the nursing discipline and person-centered care; therefore, it would be more effective to provide intensive communication education during the third year, in which students begin their clinicals, such that they can effectively communicate with patients with confidence. Moreover, students should be continuously trained for communication skills applicable in future clinical settings. 

Finally, the comprehensive PCC education program significantly reduced nursing students’ clinical practice stress compared to the control group. Nursing students have been reported to be under heavy stress during clinical practicum, primarily due to fear of unknown situations, lack of competence, and lack of control in their relationship with patients [32,33]. The program seemed to help students provide patient care with more confidence and effectively communicate with patients, thereby reducing their clinical practice stress. Previous findings that PCC increases job satisfaction while reducing stress in healthcare providers [6,7] partially support our result.

This study is significant as the first study in Korea and abroad to implement and evaluate a comprehensive PCC education program in nursing students. Moreover, the fact that we developed a feasible comprehensive PCC education program for nursing students undergoing clinical practicum adds to the significance of this study. Nursing students who completed the education program applied what they had learned into practice during on-campus and clinical practicum and evaluated the effectiveness of the program themselves, thereby improving the accuracy of the assessment. Finally, this study showed that a program that simultaneously boosts empathy and communication competence as well as PCC through theoretical and case-based education is effective, thereby establishing evidence supporting the implementation of a comprehensive intervention.

This study has a few limitations. We could not use a randomized experimental design and used a non-synchronized design to prevent any diffusion effects between the experimental group and control group. Hence, bias caused by various exogenous variables that might have occurred during this period could not be controlled. Moreover, we conducted the comprehensive PCC education program at one university in Korea; therefore, these results have limitations in generalizing the findings. Lastly, although we administered the post-test after the students had about two weeks of on-campus and clinical practicum following the completion of the intervention, we did not use a longitudinal design and thus could not evaluate whether the effects are retained in the long term. 

In the future, longitudinal randomized controlled trials are needed to substantiate the findings of this study. Moreover, replication studies are needed with nursing students from diverse universities. Finally, studies should develop more patient scenarios for case-based learning, an essential component of comprehensive PCC education, and assess the effects of such education. 

## 5. Conclusions

This study developed, implemented, and evaluated the effects of a comprehensive PCC education program in nursing students undergoing clinical practicum. The program improved nursing students’ PCC competence, empathy, and communication competence while reducing their clinical practice stress. Thus, a comprehensive PCC education program could be utilized as a pre-practicum intervention for nursing students to boost their confidence and promote PCC practice during clinical practicum. 

## Figures and Tables

**Figure 1 medicina-59-00463-f001:**
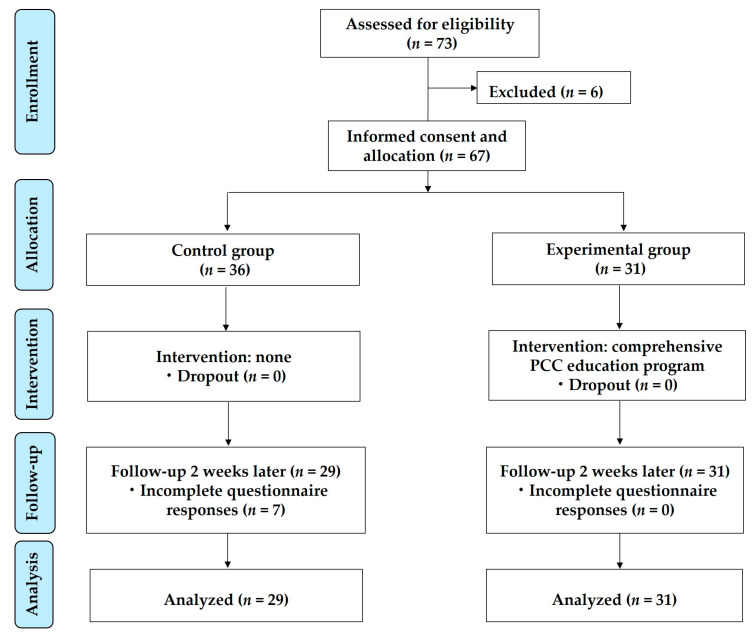
Research flow diagram.

**Table 1 medicina-59-00463-t001:** Educational contents of comprehensive person-centered care (PCC) education program.

Session	Understanding of PCC and Case Study (20 min)	Empathy Training (15 min)	Communication Training (15 min)
1	[Understanding of PCC] Definition of PCC Core premises of PCC Effects of PCC	[Concept of empathy] Concept and significance of empathy Understanding of empathy (Cases of empathy) Correct way of expressing empathy Effects of empathy expression (Practice) Expressing empathy correctly based on case study	[Asking an open-ended question] Importance and effects of open-ended questions Asking a correct open-ended question (Practice) Asking an open-ended question
2	[PCC case study 1] Case presentation Provide PCC in the case Share thoughts	[Recognizing emotion] Importance of recognizing emotion Introduce words that express emotion Recognizing my own emotions Recognizing others’ emotions (Practice) Checking off emotions felt in the past week on a list of emotional expressions and present the results	[Acknowledging] Concept and importance of acknowledgement Effects of acknowledging Correct way of acknowledging (Practice) acknowledging
3	[ PCC case study 2] Case presentation Provide PCC in the case Share thoughts	[Emotional expression] Expressing emotions Importance of correct emotional expression (Practice) Sharing thoughts after expressing emotions	[I-message] Definition of I-message Importance of I-message Effects of I-message Correct I-message (Practice) I-message
4	[ PCC case study 3] Case presentation Provide PCC in the case Share thoughts	[Listening] Concept of listening Benefits of listening Preparing for listening Correct way of effective listening (Practice) Listening	[Reflective listening] Definition and effects of reflective listening Correct way of reflective listening (Practice) Reflective listening (e.g., simple reflection, double-sided reflection)

**Table 2 medicina-59-00463-t002:** Homogeneity of general characteristics of participants between groups before intervention (n = 60).

Characteristics		Experimental Group (n = 31)	Control Group (n = 29)	χ^2^ or *t*	*p*
		n (%) or M ± SD	n (%) or M ± SD		
Sex	Male	12 (38.7)	6 (20.7)	2.317	0.128
	Female	19 (61.3)	23 (79.3)		
Age (years)		22.54 ± 1.96	22.55 ± 1.37	0.008	0.994
Religion	Yes	22 (71.0)	17 (58.6)	1.004	0.316
	No	9 (29.0)	12 (41.4)		
Subjective health status	Good	24 (77.4)	21 (72.4)	0.200	0.655
	Moderate	7 (22.6)	8 (27.6)		
Satisfaction with college life	Satisfied	21 (67.7)	18 (62.1)	0.212	0.645
	Moderate	10 (32.3)	11 (37.9)		
Satisfaction with nursing	Satisfied	21 (67.7)	17 (58.6)	2.771	0.219 ^†^
	Moderate	8 (25.8)	12 (41.4)		
	Not satisfied	2 (6.5)	0 (0.0)		
Academic performance	4.0~4.5	13 (41.9)	10 (34.5)	0.570	0.752
	3.5~3.9	12 (38.7)	14 (48.3)		
	<3.5	6 (19.4)	5 (17.2)		

^†^ Fisher’s exact test, M: mean, SD: standard deviation.

**Table 3 medicina-59-00463-t003:** Homogeneity test of the dependent variables before intervention (n = 60).

Variables	Exp. (n = 31)	Cont. (*n* = 29)	*t* or Z	*p*
	M ± SD	M ± SD		
Individualized care	59.19 ± 10.78	62.86 ± 11.09	1.29	0.173 ^‡^
Empathy	115.77 ± 7.86	112.96 ± 10.35	−1.18	0.240
Communication competence	58.54 ± 9.21	57.00 ± 8.41	−0.67	0.500
Clinical practice stress	55.38 ± 9.73	54.51 ± 11.12	−0.32	0.748

^‡^ Mann–Whitney test; Exp.: experimental group; Cont.: control group; SD: standard deviation.

**Table 4 medicina-59-00463-t004:** Comparison of dependent variables between the two groups after intervention (n = 60).

Variables	Group	Pre-Test	Post-Test	Difference (Post-Pre)	*t* or Z	*p*
		Mean ± SD	Mean ± SD	Mean ± SD		
Individualized care	Exp. (n = 31)	59.19 ± 10.78	64.80 ± 11.79	5.61 ± 11.38	−2.44	0.024 ^§^
	Cont. (n = 29)	62.86 ± 11.09	62.80 ± 10.10	−0.24 ± 6.26		
Empathy	Exp. (n = 31)	115.77 ± 7.86	119.06 ± 7.62	3.29 ± 6.90	−2.76	0.008
	Cont. (n = 29)	112.96 ± 10.35	110.58 ± 11.18	−2.37 ± 8.90		
Communication competence	Exp. (n = 31)	58.54 ± 9.21	59.87 ± 8.47	0.77 ± 4.42	−3.25	0.002
	Cont. (n = 29)	57.00 ± 8.41	54.03 ± 9.14	−2.96 ± 4.48		
Clinical practice stress	Exp. (n = 31)	55.38 ± 9.73	53.96 ± 12.14	−1.41 ± 7.93	2.90	0.005
	Cont. (n = 29)	54.51 ± 11.12	59.00 ± 12.26	4.48 ± 7.81		

^§^ Mann–Whitney test; Exp.: experimental group; Cont.: control group; SD: standard deviation.

## Data Availability

The study data are available upon request from the corresponding author.

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
