# Peer review of "Effects of a Comprehensive Person-Centered Care Education Program for Nursing Students"

_medicina, 2023, doi:10.3390/medicina59030463_

Round 1
Reviewer 1 Report
Thank you very much for giving me the opportunity to review this manuscript. The author presents the results of an educational programme focused on person-centred care in third year nursing students. I believe that the concepts you address to try to provide such care are appropriate, however I have some questions that I would like you to answer, and that perhaps you should include in your manuscript.
What is the situation of nursing students in your university? That is, have these students done clinical practice in a hospital before? This may have an influence.
How do you assess empathy and communication activities? That is, if the skills are performed adequately, how do you assess whether a student understands the case study or not? I think this is important for you to make clear in your manuscript.
I think the intervention you have implemented needs to be described more in terms of content and form
Author Response
I greatly appreciate your thoughtful comments and incorporated your suggestions. Please find my responses below. I highlighted the revised parts in yellow in my updated manuscript.

Reviewer 2 Report
The article raises an interesting question and some research justifications are included.
Lines 63-69. You refer that no study has yet reported the effects of a comprehensive PCC education program, but if we disregard effects, are there similar education programs that could be used? this should be included in a reflection as one can learn from similar programs
Otherwise method and results, discussion and conclusion are well written with many considerations
Author Response

(The authors gave the same response as above.)
